FV-EffResNet: an efficient lightweight convolutional neural network for finger vein recognition

Tahir Yusuf Suleiman
Rosdi Bakhtiar Affendi eebakhtiar@usm.my
School of Electrical and Electronic Engineering, Universiti Sains Malaysia , Nibong Tebal , Penang , Malaysia
Sadek Rowayda
Electronic publication date: 2024 Feb 15
Publication date: 2024
Volume: 10
Electronic Location ID: e1837
Received 2023 Aug 23; Accepted 2024 Jan 7
Copyright: ©2024 Tahir and Rosdi
Copyright year: 2024
Copyright holder: Tahir and Rosdi
License: This is an open access article distributed under the terms of the Creative Commons Attribution License, which permits unrestricted use, distribution, reproduction and adaptation in any medium and for any purpose provided that it is properly attributed. For attribution, the original author(s), title, publication source (PeerJ Computer Science) and either DOI or URL of the article must be cited.
License URL: https://creativecommons.org/licenses/by/4.0/

Keywords: Biometrics, Deep learning, Convolutional neural network, Cyclical learning, Lightweight network, Swish, EffRes block, Finger vein recognition

Funding: The Ministry of Higher Education Malaysia for Fundamental Research Grant Scheme with Project Code FRGS/1/2021/ICT07/USM/02/2 This work was supported by the Ministry of Higher Education Malaysia for Fundamental Research Grant Scheme with Project Code: FRGS/1/2021/ICT07/USM/02/2. The funders had no role in study design, data collection and analysis, decision to publish, or preparation of the manuscript.

==============================
Several deep neural networks have been introduced for finger vein recognition over time, and these networks have demonstrated high levels of performance. However, most current state-of-the-art deep learning systems use networks with increasing layers and parameters, resulting in greater computational costs and complexity. This can make them impractical for real-time implementation, particularly on embedded hardware. To address these challenges, this article concentrates on developing a lightweight convolutional neural network (CNN) named FV-EffResNet for finger vein recognition, aiming to find a balance between network size, speed, and accuracy. The key improvement lies in the utilization of the proposed novel convolution block named the Efficient Residual (EffRes) block, crafted to facilitate efficient feature extraction while minimizing the parameter count. The block decomposes the convolution process, employing pointwise and depthwise convolutions with a specific rectangular dimension realized in two layers (n × 1) and (1 × m) for enhanced handling of finger vein data. The approach achieves computational efficiency through a combination of squeeze units, depthwise convolution, and a pooling strategy. The hidden layers of the network use the Swish activation function, which has been shown to enhance performance compared to conventional functions like ReLU or Leaky ReLU. Furthermore, the article adopts cyclical learning rate techniques to expedite the training process of the proposed network. The effectiveness of the proposed pipeline is demonstrated through comprehensive experiments conducted on four benchmark databases, namely FV-USM, SDUMLA, MMCBNU_600, and NUPT-FV. The experimental results reveal that the EffRes block has a remarkable impact on finger vein recognition. The proposed FV-EffResNet achieves state-of-the-art performance in both identification and verification settings, leveraging the benefits of being lightweight and incurring low computational costs.

Introduction

Biometric recognition is the process of identifying people based on some biological traits that are either extrinsic or intrinsic. Fingerprints (Nahar, Chaudhari & Tanwani, 2022; Deshpande et al., 2020; Abrishambaf, Demirel & Kale, 2008), faces (Bah & Ming, 2020; Pranav & Manikandan, 2020; Hurieh, Mansouri & Mohammad, 2014; Lenc & Král, 2015), and irises (Mahameed, Ahmed & Abdullah, 2022; Thomas, George & Devi, 2016) are some of the frequently utilized extrinsic features in this regard. However, each of these modalities has its limitations. For example, the age and occupation of a person may cause some difficulties in capturing a complete and accurate fingerprint image (Galbally, Haraksim & Beslay, 2018), and iris scans may be uncomfortable for some users (Omidiora et al., 2015). Also, these traits are vulnerable to spoofing attempts in an unattended environment (Alzubaidi et al., 2021). On the other hand, intrinsic modalities are hidden beneath the skin. As such, they are more difficult to fake (Shaheed et al., 2018). Among the modalities in this category is the finger vein. Recently, they have attracted a lot of research as a novel biometric method (Shaheed et al., 2022; Tamang & Kim, 2022; Chai et al., 2022; Song, Kim & Park, 2019; Lu, Xie & Wu, 2019; Hong, Lee & Park, 2017). This is mainly due to their uniqueness to each individual, high security, and insulation against disease or accidents.

The finger vein images are mostly acquired using near-infrared imaging techniques. However, these images are generally of low quality and include not only vein patterns but also irregular shading and noise due to un-even illuminations, varying tissues, and bones, changes in the physical conditions, and blood flow making the thickness and brightness of the same vein different in each acquisition. Therefore, there is a crucial need for a robust feature extraction technique for the accurate representation of these veins (Choi et al., 2009).

Conventionally, the finger vein recognition pipeline includes image acquisition and preprocessing, feature extraction, and matching. This type of approach is generally referred to as hand-crafted or feature engineering methods. Over the years, researchers have proposed several feature engineering methods to represent the appearance of finger veins. These methods are broadly classified into three groups: methods based on vein pattern, local binary-based methods, and methods based on dimensionality reduction (Université Yahia Fares de Médéa and Institute of Electrical and Electronics Engineers, 2018).

In the vein-pattern based group, the geometric features or topological structure of the vein pattern are firstly segmented and then used as templates for matching. Repeated line tracking (Miura, Nagasaka & Miyatake, 2004), modified repeated line tracking (Liu et al., 2013), Gabor filters (Jinfeng, Shi & Jinli, 2010; Khellat-kihel et al., 2014; Peng et al., 2012), maximum curvature (Syarif et al. 2017; Li et al., 2019) and mean curvature (Song et al., 2011; Yang et al., 2014) all belong in this group.

The local binary pattern-based methods are based on the local area, an image is divided into small, non-overlapping cells, and the texture patterns within each cell are analyzed. The method computes a binary code for each pixel within a cell, based on its relationship with its surrounding pixels. The code is then concatenated to form a single feature vector that describes the texture patterns within the cell. Methods utilized for finger veins in this category include local line binary pattern (LLBP) (Rosdi, Shing & Suandi, 2011), local binary pattern (LBP) (Wang et al. 2012; Hu, Ma & Zhan, 2018), personalized best bit map (PBBM) (Yang, Xi & Yin, 2012), local directional code (LDC) (Meng et al., 2012) and personalized weight maps (PWM) (Yang et al., 2013).

Dimensionality reduction-based methods use machine learning algorithms to transform the images into a lower dimensional space by projecting them onto a new set of variables that capture the most important features. These features are then easily used for classification. PCA (Wu & Liu, 2011; Yang, Xi & Yin, 2012) and its variants are some examples of the dimensionality reduction-based methods proposed for finger vein recognition.

Although feature engineering methods are shown to be effective in the finger vein recognition domain, most of these methods require researchers to have some degree of expertise about vein patterns and are therefore not very robust. Furthermore, they tend to be computationally expensive which limits their feasibility for real-time applications (Sidiropoulos et al., 2021). Deep learning (DL) methods have recently attracted much research attention owing to their superior performance in image-processing tasks like object detection and classification (Ponti et al., 2017). Their effectiveness in extracting features from images and their end-to-end nature have prompted their introduction in the finger vein biometrics domain. It has been shown to automatically extract more specialized and distinctive features from finger vein images relative to hand-crafted methods. It is worth noting that proposed methods have been evaluated in both identification and verification settings, exploring their potential for accurate and reliable recognition in the context of finger vein biometrics

Among the seminal works utilizing DL for finger vein recognition evaluated in identification settings is the system proposed by Das et al. (2018). The proposed CNN-based network takes non-square images as input to avoid severe resizing operations that could negatively impact the information in the images. This study was the first to evaluate its model on four different publicly available databases, achieving stable and high accuracy across all of them. Gumusbas et al. (2019) in their work leveraged capsule networks for their advantages in rotational and translational invariance to extract robust finger vein features for identification. In contrast to CNNs, capsule networks utilize vector representations of features stored in capsule layers instead of numerical activation values. These vector representations encode both the length and orientation information of the vector, allowing for richer and more informative representations of features. In similar research (Boucherit et al., 2022), an architecture consisting of two identical short-path CNNs is proposed. Each takes as input the same finger vein image with different quality, their outputs are then unified into a single one using the late fusion strategy. However, the late fusion strategy requires more computational time because there are more classifiers to train. Inspired by the shape of finger vein images Chai et al. (2022) in their work introduced a lightweight network by merging lower pre-trained layers of MobileNet with newly constructed layers. These new layers were initialized with rectangular convolution kernels, demonstrating slightly better accuracy compared to previous finger vein recognition networks using square kernels. In Tamang & Kim (2022), the process of feature extraction is enhanced through hybrid pooling, which involves the placement of max-pooling and average pooling in parallel after each convolutional layer and their outputs are concatenated before passing to another convolutional layer. This arrangement allows the network to extract multi-level and discrete interclass features of the input. However, the proposed pipeline not only has a large number of parameters but also includes vigorous preprocessing of the input thereby making the proposed work complex. In Liu et al. (2022), the proposed fusion residual attention block includes a main path that extracts features at multiple scales and a guided attention path corresponding to the main path feature map. Venous features extracted at different learning stages through these two pathways are integrated using a multistage residual attention scheme. Ma, Wang & Hu (2023) devised a multiple attention module specifically tailored for the finger vein. This module was integrated with mixed depth-wise convolutions utilizing different kernel sizes to build a network named multi-scale feature bilinear fusion network which simultaneously extracts channel, spatial, directional, and positional information from the finger vein.

Conversely, some of the methods evaluated in verification settings include the work presented in Tang et al. (2019), which introduced a novel method based on siamese CNN to enhance finger vein verification requirements. The proposed approach integrates mini-ROIs and the original vein images through a two-stream network. Multi-task learning approaches have also been integrated into the realm of finger vein recognition. In Hao, Fang & Yang (2020), the authors employed a hard-sharing method to create a multi-task network model with two dedicated branches for region of interest (ROI) extraction and finger vein feature extraction. The initial step involved the ROI branch generating parameters for the specified region. These parameters were then used as inputs for the finger vein classification branch, enabling classification based on values derived from the proposed region on the original feature maps. Furthermore, Yang et al. (2020) proposed a joint network that simultaneously handles the tasks of recognition and anti-spoofing. The suggested approach exhibited remarkable results across both evaluation metrics, highlighting the effectiveness of multi-task learning in the finger vein domain. A multimodal approach incorporating features from both finger vein and finger shape modalities is proposed by Kim, Song & Park (2018). Simultaneous extraction of features from the two modalities is achieved using two pre-trained ResNets. Matching distances calculated based on the features of finger vein and finger shape obtained using the ResNet models are score fused using various fusion methods. These fusion methods include the weighted sum, weighted product, and perceptron to determine the most efficient approach. In another research Lu, Xie & Wu (2019), a pre-trained model was employed to create a CNN-based local descriptor named CNN Competitive Order (CNN-CO). The outputs of chosen CNN filters were utilized to generate CNN-CO, providing a discriminative representation of finger vein images. The proposed local descriptor was tested on two public databases, demonstrating a low equal error rate (EER) in both cases. Lin et al. (2021) proposed a novel approach to enhance finger vein recognition by leveraging both intrinsic and extrinsic features of finger vein images. The methodology centers around two crucial modules: an intrinsic feature learning module designed to estimate the expectation of intra-class finger vein images, accommodating various offsets and rotations, and an extrinsic feature learning module to learn inter-class feature representations. Robust verification is achieved by evaluating the distances between both intrinsic and extrinsic features. Huang et al. (2021), introduced the joint attention (JA) module. This attention mechanism leverages the interdependence among channels and spatial locations within feature maps to enhance the focus on vein patterns. To assess the efficacy of the proposed JA module, the authors constructed a shallow network. Within this network, a generalized mean pooling layer was embedded to obtain a concise feature descriptor. In contrast to prior DL approaches for finger vein verification, the research outlined in Huang et al. (2023) explored deep feature extraction within the frequency domain. The study introduces a novel frequency convolution module and demonstrates its synergistic integration with spatial convolution, yielding outstanding results in finger vein authentication. Furthermore, the research offers a comprehensive set of experimental results, utilizing nine readily available finger vein datasets.

Although previous DL models have an outstanding performance for finger vein biometrics, they still exhibit certain drawbacks that constrain their real-time implementation, especially on embedded hardware with limited resources. These drawbacks include the use of extensive preprocessing on the input images. Additionally, many deep learning-based methods encounter challenges when dealing with small-scale finger vein datasets due to their large and complex architectures, which often result in longer training and inference times and also make them susceptible to overfitting. It is also worth noting that most of the DL methods described above adopted the conventional ReLU function as the activation in the hidden layers of their proposed networks mainly due to its popularity among ImageNet models while ignoring the possibility of one more suited for finger vein models.

To alleviate the above-mentioned drawbacks, this article proposed a lightweight architecture called FV-EffResNet based on an Efficient Residual (EffRes) block. Specifically, the EffRes block is crafted with the dual objective of optimizing data flow for faster computations and ensuring robust feature extraction capabilities on finger vein data. The proposed method eliminates the need for intricate processing of the original images, such as feature enhancement, and substantially reduces the complexity of the method. The primary contributions of this article are as follows:

• We introduced a novel convolution block named EffRes block, specifically designed for efficient feature extraction while minimizing parameter count. The block decomposes the convolution process, employing pointwise and depthwise convolutions realized through two consecutive layers with n × 1 and 1 × m convolutions, as opposed to the conventional n × m convolution. Notably, a rectangular dimension (3 × 7 decomposed into 3 × 1 and 1 × 7) is employed, which is found to be more suitable for finger vein data. Residual connections over each block are included to improve gradient flow.

• A new finger vein recognition architecture, namely, FV-EffResNet. The architecture is characterized by its shallow and lightweight design with only 1.32M parameters achieved through the stacking of EffRes blocks. The network incorporates Swish activation (Ramachandran, Zoph, & Le, 2017) in hidden layers, hypothesized to be advantageous for finger vein networks due to its unique properties. Additionally, the adoption of the cyclic learning rate technique (Smith, 2017) is employed to expedite network convergence and mitigate overfitting.

• The positive impact of the EffRes block and the efficacy of FV-EffResNet are extensively examined through a comprehensive set of comparative experiments in both identification and verification settings. The evaluations are conducted on FV-USM, MMCBNU_600, SDUMLA, and NUPT-FV databases.

The rest of the article is organized as follows: in ‘Materials & Methods’, the proposed method is described in detail. ‘Experiments and Analysis’ presents an ablation study of the proposed network, and we present a comparison of our method with recent state-of-the-art methods in ‘Comparison with State of the Art’. Finally, conclusions in ‘Conclusions’.

Materials & Methods

In this section, we present a novel approach for finger vein recognition based on CNN and its motivation. We also thoroughly examine the attributes of the datasets utilized in training our proposed method. The overall pipeline includes two stages: the network design and training stage. For clarity, we will initially introduce some prerequisite background about the relevant methods used, and then proceed to describe our proposed approach.

Background

Convolution block

The core component of deep CNNs is the convolutional layer. A standard convolutional layer is designed to extract features from an input three-dimensional volume, comprising spatial dimensions (width and height) and a depth/channel dimension. As a result, each kernel in the convolutional layer is tasked with simultaneously learning mappings for cross-channel correlations and spatial correlations. Executing this operation with multiple kernels involves quadruple-component multiplication resulting in quadratic complexity (Chollet, 2017; Freeman, Roese-Koerner & Kummert, 2018), which comes with nontrivial computational cost. A common trend observed in several early architectures for finger vein recognition (Das et al., 2018; Boucherit et al., 2022; Gumusbas et al., 2019; Tamang & Kim, 2022; Yang et al. 2020; Hao, Fang & Yang, 2020) is the performance of feature extraction through multiple successive layers without any form of spatial downscaling. This approach leads to networks with a significant number of parameters and, consequently, a high computational cost. In more recent works, there has been a shift towards utilizing various types of optimized convolutional layers to give rise to lightweight finger vein recognition networks (Liu et al., 2022; Chai et al., 2022; Yang et al., 2022). Our primary contribution in this work is also in this aspect of optimization through (re)design of the convolutional blocks to enhance network performance and reduce parameter count. A detailed description of our proposed method is presented in a subsequent section.

Activation function

Activation functions are an essential part of neural networks. An activation function is a mathematical function that takes in the weighted sum of inputs to a neuron, applies a non-linear transformation to it, and outputs the results, which are then used as input for the subsequent layer in the network (Sharma, Sharma & Athaiya, 2020). The purpose of an activation function is to add non-linearity to the output neuron. Without an activation function, the output of a neuron would simply be a linear function of its input, regardless of the number of layers in the network. Adding non-linearity to the output of a neuron allows a neural network to learn more complex functions and better represent non-linear relationships in the data it is trained on. The hidden layers of a neural network employ a uniform activation function, while the output layer’s activation function is typically distinct and relies on the model’s prediction objectives. Typically, differentiable non-linear activation functions are used in the hidden layers of a neural network. This allows the backpropagation strategy to be utilized for calculating weight losses and ultimately optimizing the weights through gradient descent or other optimization techniques to minimize errors (Ponti et al., 2017). Among the most popular activation functions proposed is the rectified linear unit (ReLU) (Nair & Hinton, 2010). ReLU is the most widely adopted activation in hidden layers of deep neural networks due to its simplicity and is also effective in addressing the problem of vanishing gradients, which is a common issue in deep networks. However, in a more recent work (Ramachandran, Zoph & Le, 2017), a non-linear function that combines a linear component with a sigmoid activation named Swish was proposed. The Swish function is a smooth and continuous function, unlike the ReLU which is a piecewise linear function. It allows a small number of negative weights to be propagated through, which has been shown to have a significant improvement on the performance of the system over the conventional ReLU, especially with deeper networks or challenging datasets and can be easily incorporated into deep networks using backpropagation. The Swish function is represented mathematically by Eq. (1), with its plot shown graphically in Fig. 1. (1) fx=x.sigmoidβx.βdenotesalearnableparameter.

Figure 1 The Swish activation function.

Cyclical learning rate

Learning rate (LR) is a crucial hyperparameter in the training of neural networks, as it plays a vital role in determining the speed and efficacy of the training process. Essentially, the learning rate determines the extent to which the loss gradient will be applied to the current weights, guiding them toward a lower loss. In a prior work Smith (2017), described a technique for setting and tweaking learning rate during the training of a neural network named cyclical learning rate (CLR). The basic idea of CLR is to vary the learning rate cyclically during training, which has been shown to help a network converge faster and to achieve improved classification accuracy of a network without a need to tune and often in fewer iterations. The cycle is defined by two hyperparameters: the maximum and minimum learning rate and the step size which is the number of iterations it takes to complete a half cycle. In a complete cycle, the learning process starts with a small LR and gradually increases over a few epochs until it reaches a maximum value, then decreases back to the minimum value. This cycle is repeated until the end of the training process. The logic behind this technique is that periodic higher learning rates within each epoch help the network come out of any saddle points or local minima if it encounters one during training. The CLR technique also described three mode policies: triangular, triangular2, and exp, for specifying the pattern in which the LR varies within the maximum and minimum bounds. The maximum and minimum LR values are estimated by conducting an “LR range test”, where the network is initially run for a few epochs during which the LR is increased linearly within a large boundary. The values of when the accuracy starts to increase and when it starts to fall are taken to be the max and min LR value rates. The LR test is of huge benefit especially when dealing with a new architecture or dataset as it allows for the estimation of the initial learning rate, which practically eliminates the need to experimentally find the best values and schedule for the global learning.

Finger vein ROI extraction

Finger vein ROI extraction involves isolating and clipping out the finger region with the most abundant vein pattern and eliminating undesirable areas from the raw images. In this work, we utilized the method proposed by Lu et al. (2021), which is based on the characteristics of the original finger vein image. Their algorithm starts by searching for the finger edges based on the finger contour imaging characteristics. This method combines the original vein image with the improved edge detection operator to achieve accurate detection of the finger edges. In the case of rotations, correction angles are calculated using least square estimation and corrected by affine transformation. The joint cavity closest to the tip of the finger is then located and used as a reference. Finally, the ROI images are segmented from the original images using the detected edges and the joint cavity as boundaries. It is noteworthy to state that our method does not require any additional preprocessing or enhancements other than standard ROI extraction which is also used by all other competing models as it is essential in eliminating undesirable areas.

Proposed network

EffRes block

The basic idea behind the proposed EffRes block is to enhance the efficiency of feature extraction while simultaneously minimizing the parameter count. This is accomplished by explicitly decomposing the convolution process into a sequence of operations. Specifically, the block begins similarly to Iandola et al. (2016) with a pointwise convolution, aiming to distill relevant information from inter-channel relationships in the input data. Simultaneously, it functions as a squeezing unit, reducing the number of input filters to the next layer by half.

The output of the preceding operation then undergoes depthwise convolution, realized by two consecutive layers using n × 1 and 1 × m convolutions respectively, rather than the conventional n × m convolution. Notably, a rectangular dimension (3 × 7 decomposed into 3 × 1 and 1 × 7) is employed, which is found to be more suitable for finger vein data, as indicated in Chai et al. (2022). A key distinction lies in our proposed rectangular convolution block kernel, which is implemented through two consecutive layers, deviating from Chai et al. (2022), where the conventional convolution layers with a (3 × 9) kernel are used. Both layers include a nonlinear activation followed by a batch norm layer. We adopt depthwise convolution to significantly reduce the number of computations compared to standard convolution, with further decomposition into two consecutive layers providing additional computational efficiency. The latter operation alone does not drastically reduce the computational complexity. However, when combined with the pooling strategy introduced by Freeman, Roese-Koerner & Kummert (2018) (which performs pooling after the first spatial layer), significant computational savings are achieved from the second layer and subsequent layers, contributing to an overall reduction in computation cost.

Each block being converted into an EffRes block may or may not contain a subsampling operation. In the former case, we split the subsampling operation into two operations: the first will be performed (via a 1 ×2 max-pooling operation) after the first/initial depthwise convolution, while the other is realized using a strided convolution (with a 1 ×2 kernel) after the last depthwise convolution in the block. In the latter case (i.e., no subsampling), max pooling is removed, and the last convolution is realized using a pointwise convolution. We also include residual connections over each block—from before the initial 1 × 1 convolution to just after the final 1 × 1 output—to improve gradient flow. The overall architecture of the proposed EffRes block both with and without subsampling is shown in Fig. 2.

Overall architecture

The overall architecture of the proposed model is illustrated in Fig. 3. The EffRes blocks described above constitute the most critical components of the entire model. FV-EffResNet is deliberately designed to be lightweight. The architectural design follows the classical deep-narrow approach to enhance multi-level feature extraction capability and alleviate the computational burden. As depicted in Fig. 3. as the network deepens, spatial dimensions’ decrease, and channel dimensions gradually increase. The network receives extracted ROIs with dimensions of (64 × 192 × 3) as input. A sequence of the EffRes blocks progressively transforms the input, resulting in a feature map of dimensions (2 × 6). This transformed feature map is then fed to the classification block. The classification block itself consists of an adaptive average pooling layer, followed by a flatten layer, and concluded with a softmax layer which yields probability values for each subject, facilitating classification.

Figure 2 The proposed EffRes block (A) with subsampling (B) without subsampling.

Figure 3 The overall architecture of the proposed FV-EffResNet.

Network training strategy

The CLR technique has been adopted for training our proposed network. As described earlier, the CLR approach involves cycling the learning rate between a minimum and maximum value in a specified pattern during the training process. The triangular window pattern as illustrated in, Fig. 4, is selected as the policy for training with the proposed model. With a triangular policy, the learning rate monotonically increases to the maximum learning rate from the base learning rate and decreases back to the base learning rate in a triangular manner. The first step in applying CLR is the “LR range test” where the maximum learning rate and the base learning rate are estimated (Smith, 2017). This test involves picking very low LR and gradually increasing the learning rate during a short training process of typically three to five cycles while monitoring the loss function. At each step, the loss function is recorded, and the learning rate is increased by a small amount. The process is repeated until the loss function starts to increase significantly, indicating that the learning rate is too high. Using the proposed model, four separate LR range tests on each of the used databases are conducted.

Figure 4 The triangular policy of cyclical learning rate.

Based on the number of samples in each dataset, the cycle length and step size are computed from the number of iterations in an epoch. An epoch is calculated by dividing the number of training images by the batch size used. As noted in Smith (2017), experiments show that it is optimal to set step size in the range of two to ten times the number of iterations in an epoch. The optimal step size for training our model with all datasets is eight times the number of iterations in an epoch. This was obtained by experimenting with different step size values. Also, overall training is stopped at the end of a cycle, which is when the learning rate is at the minimum value and the accuracy peaks.

Finger vein databases

FV-USM Database (Asaari et al., 2014): Developed at the Universiti Sains Malaysia, it consists of finger vein images of 123 subjects acquired in two sessions. All subjects participated in both sessions and images from four fingers were captured for each subject: left index, left middle, right index, and right middle fingers. For each finger, six samples were taken. Thus, a total of 2,952 (123 × 4 × 6) images were captured in each session making an overall total of 5,904 images from 492 finger classes. In the database, ROI images of the finger vein which were extracted based on the method presented in Asaari et al. (2014) are also provided.

MMCBNU_600 (Lu et al., 2013): Conducted by Chonbuk National University in Jeonju, South Korea, comprises images from 100 individuals, capturing the index, middle, and ring fingers of both hands. Each finger contributes ten images, resulting in a total of 6,000 images. All raw images are grayscale and possess a resolution of 640  × 480. Additionally, the dataset includes the extracted ROI with a resolution of 128  × 60.

SDUMLA-HMT (Yin, Liu & Sun, 2011): This database was collected by Shandong University’s group of Machine Learning and Applications and presented in their research (Yin, Liu & Sun, 2011) It consists of finger vein images from 106 individuals. Six samples from the left and right hand’s index, middle, and ring fingers were acquired from each of the individuals. All images were acquired in a single session which summed up to a total of 3,816 images from 636 finger classes

NUPT-FPV(Ren et al., 2022): It contains a total of 16,800 finger vein images collected from 140 subjects in two sessions. For each subject, the images of six different fingers were captured, 10 samples for each finger. All subjects which include 108 males and 32 females participated in both sessions. The database was collected by the Nanjing University of Posts and Telecommunications (NJUPT).

For the FV-USM, MMCBNU_600, and NUPT-FPV we used the database’s given ROIs as the input for the proposed model. We adopted the ROI extraction method described in section II for the SDUMLA dataset. The method is based on the characteristics of the original finger vein image. Before training the proposed network, all the finger vein ROI images are uniformly resized to 64 × 192 thereby maintaining the aspect ratio of the original ROIs to avoid severe distortion. The characteristics of the employed dataset are summarized in Table 1.

Table 1 A summary of the considered finger vein database.

Databases	Subjects	No fingers per subject	No. samples per finger	No. of sessions	Image size	Image format	Total no. of images	
FV-USM	123	4	6	2	640 × 480	JPEG	5,904	
SDUMLA	106	6	6	1	320 × 240	BMP	3,816	
MMCBNU_600	100	6	10	1	640 × 480	BMP	6,000	
NUPT-FPV	140	6	10	2	300 × 400	BMP	16,800	

Experiments and Analysis

Experimental setup

Experiments were conducted using Python programming language with pytorch framework running on a high-performance desktop with Nvidia RTX A5000, 24 GB GPU. The batch size used for training is set to 8, and Adam is used as the optimizer, with weight-decay set to 0.0001. The cyclicLR schedule, detailed in the previous section, is employed as the learning rate scheduler. In experiments where the cyclicLR scheduler is used, the step size up is set to be eight times the number of iterations, and the step size down is set to two times the step size up. A summary of the parametric settings is presented in Table 2 which are kept the same in all experiments. Each finger of a subject is considered a separate class. To validate the effectiveness and contributions of each component of our proposed FV-EffResNet, ablation experiments are presented in this section. The first experiment deals with verifying the efficacy of using Swish activation in finger vein networks over the conventional ReLU and its variant. In the second experiment, the capability of the proposed EffRes block to extract robust features is studied. Finally, in the last experiment, we investigated the effectiveness of using the cyclic learning rate schedule for faster network training (convergence).

Table 2 Parametric setting used in training of FV-EffResNet.

Parameters	Values	
Batch size	8	
Optimizer	Adam	
Loss function	cross entropy	
Weight decay	0.0001	
Learning rate scheduler	cyclicLR	
Step up size	8 * Total number of iteration in an epoch	
Step down size	2 * Step up size	

Furthermore, to ensure a fair comparison with other methods, the proposed network is trained on each considered database using different train-test split ratios that match those employed in other methods. After completing the model training, it can extract finger vein features and provide the probability values for the test image belonging to the known classes. The class associated with the highest probability value is considered the predicted result of the CNN model for the test image. We assess the effectiveness of our proposed method in both identification and verification settings and have employed a closed-set evaluation approach in each case. This decision was made to facilitate a more effective and meaningful comparison with the existing literature in the field of finger vein recognition, as the majority of these studies inherently utilize closed-set evaluation methodologies. All results are presented and analyzed in the subsequent sub-sections.

Experiment 1: Impact of Swish activation function

To investigate the efficacy of the Swish function with finger vein datasets, we developed a baseline CNN model that comprises six blocks for feature extraction and an output block. Each block consists of a normal convolution layer with rectangular kernels, a non-linear layer, and a batch normalization layer. The first five feature extraction blocks use a kernel size of (3 × 7) in the convolution layers, while the last uses (2 × 6) thereby converting the feature map size to (1 × 1). Finally, the output block consists of adaptive average pooling, flatten layer, and softmax layer which gives a range of probability values for the number of finger classes. The number of channels in the first block was chosen at random which doubles after each block. To reduce the computational cost, subsampling layers (max pooling) are placed after each block except the last. The Swish function and two other conventional functions; ReLU and LeakyReLU were each used with our baseline model. To ensure a fair comparison, the network was trained using the Adam optimizer with the default learning rate of 0.001 for 50 epochs on each dataset. As seen from the results presented in Table 3, utilizing the Swish function as the activation in the hidden layers of the network gave slightly better validation accuracy with all five datasets, so it was adopted in all subsequent experiments.

Table 3 Training accuracy comparison of baseline model using ReLU, LeakyReLU and Swish activation function.

Activation function	FV-USM	SDUMLA	MMCBNU_600	NUPT-FPV	
ReLU	89.210%	90.07%	92.12%	92.63%	
LeakyReLU	92.11%	92.87%	92.94%	91.78%	
Swish	93.36%	93.89%	94.21%	92.81%	

Experiment 2: Impact of using EffRes blocks

The second experiment in our network design deals with optimizing our baseline model from the first experiment to a much lighter variant without sacrificing accuracy. To achieve this, we scaled down the network by replacing its blocks with the proposed EffRes blocks resulting in a much lighter and faster network with only 1.32 M parameters compared to the baseline model with 5.38 M parameters. Here, both versions are trained fully using a fixed LR, and early stopping callback was employed to halt training just before the network overfits. In Table 4, we present and compare the rank-1 accuracy achieved by both the initial baseline model and the scaled-down version named FV-EffResNet on the benchmark datasets. Notably, FV-EffResNet achieved convergence in only forty-eight epochs for FV_USM, forty epochs for SDUMLA, forty-three for MMCBNU_600, and fifty-three for NUPT-FV which are almost twice as fast compared to the baseline model, which the minimum was with MMCBNU_600 converging at eighty-six epochs. It is important to note that the results presented in Table 4 are based on one of the train-test splits used in evaluating the network. Specifically, for FV_USM, we utilized the first session for training and the second session for testing, while for the rest of the employed databases, we equally divided the data between training and testing. Despite FV-EffResNet’s accuracy being marginally lower than the baseline model, this is expected due to the significant reduction in the number of parameters.

Table 4 Rank-1 accuracy comparison of proposed FV-EffResNet and baseline model.

	FV-USM	SDUMLA	MMCBNU_600	NUPT-FPV	
Baseline model	95.39%	96.89%	97.51%	96.23%	
Proposed FV-EffResNet	94.14%	95.22%	97.00%	96.03%	

Experiment 3: Impact of cyclic learning schedule training strategy

In this section, we demonstrate the impact of employing the CLR learning technique. The proposed FV-EffResNet underwent training on all four finger vein datasets, utilizing the same train-test data split strategy described in the previous experiment, with training conducted using both the fixed learning rate and the CLR method. The results are compared based on the number of epochs required for the network to fully converge in both cases and the test accuracy achieved with each method. To train the network using the fixed learning rate method, LR values were determined through a trial-and-error approach. The default Adam LR value, which is 0.001, was found to be suboptimal for SDUMLA and MMCBNU_600, and an LR value of 0.003 was chosen for FV-USM and NUPT-FPV, respectively. These LR values yielded near-optimal results during the network training process for the respective datasets. With the CLR method, learning rate boundaries were first estimated through the “LR Range Test” as described in the ‘Materials & Methods’ section. The triangular policy is employed with the learning rate parameters throughout the entire network training. As illustrated in, Table 5, training the network using the CLR method on all four datasets resulted in enhanced accuracy performance and faster convergence in all the experiments. Although some learning rate values within the range led to performance drops during training, the overall results affirm the advantages of utilizing the CLR method for faster network training.

Table 5 Comparison between fixed learning rate method and cyclical learning rate method with FV-EffResNet.

Learning method	Database	No of epochs	Accuracy	
Fixed LR	FV-USM	48	94.14%	
	SDUMLA	40	92.73%	
	MMCBNU_600	43	93.50%	
	NUPT-FPV	53	93.76%	
Cyclical LR	FV-USM	28	95.75%	
	SDUMLA	24	96.11%	
	MMCBNU_600	24	98.43%	
	NUPT-FPV	32	96.84%	

Comparison with State of the Art

Finger vein identification

Finger vein identification involves the comparison of a query image with a set of training images to assign it to a class label. Rank-1 accuracy and the cumulative match characteristic curve (CMC) are commonly employed to evaluate the identification performance of a biometric model. The comparison of Rank-1 accuracy results of FV-EffResNet and other methods is presented in, Tables 6, 7, 8 and 9. Figure 5 shows the CMC curves for the comparison across each considered database, utilizing different train-test split ratios. As shown in Table 6, for the FV_USM dataset, we conducted various comparisons based on the number of sessions utilized and the train-test split ratio. Specifically, when only the first session was employed, as in Boucherit et al. (2022) and Tamang & Kim (2022), for each subject, they utilized four images for training and two for testing. Employing the same configuration with our method resulted in superior performance. When both the first and second sessions were utilized, the compared methods employed different strategies. Boucherit et al. (2022), Chai et al. (2022) utilized the first session for training and tested with second session data. Adopting a similar approach, our method achieved a significant increase compared to Boucherit et al. (2022), with an almost 15% increase in accuracy. When data from both sessions are mixed, as in Das et al. (2018), Liu et al. (2022), Ma, Wang & Hu (2023) configuration, our method achieves similar results.

Table 6 Rank-1 accuracy comparison of proposed method with other methods using two sessions of FV_USM dataset.

Train/Test split	Method	Rank-1 accuracy (%)	
Training 4 Testing 2	Boucherit et al. (2022) *	96.15	
Tamang & Kim (2022) *	97.22	
FV-EffResNet*	97.56	
Training 6 Testing 6	Boucherit et al. (2022) **	81.71	
Chai et al. (2022) **	94.67	
Das et al. (2018)	97.53	
FV-EffResNet**	95.75	
FV-EffResNet	98.79	
Training 8 Testing 4	Liu et al. (2022)	99.75	
FV-EffResNet	99.54	
Training 10 Testing 2	Ma, Wang & Hu (2023)	99.90	
	FV-EffResNet	99.39	
Notes.

* Training and Testing with only first session.

** Training with first session and Testing with second session.

Results with no superscript indicate first and second session combined.

Table 7 Rank-1 Accuracy comparison of proposed FV-EffResNet with other methods using MMCBNU_600 dataset.

Train/Test Split	Method	Rank-1 Accuracy (%)	
Training 5 Testing 5	Gumusbas et al. (2019)	95.5	
FV-EffResNet	98.43	
Training 6 Testing 4	Gumusbas et al. (2019)	95.80	
Liu et al. (2022)	99.83	
FV-EffResNet	98.58	
Training 8 Testing 2	Gumusbas et al. (2019)	100	
FV-EffResNet	99.25	

Table 8 Rank-1 Accuracy comparison of proposed FV-EffResNet with other methods using SDUMLA dataset.

Train/Test split	Method	Accuracy (%)	
Training 3 Testing 3	Gumusbas et al. (2019)	87.00	
Chai et al. (2022)	96.61	
FV-EffResNet	95.75	
Training 4 Testing 2	Das et al. (2018)	97.48	
Gumusbas et al. (2019)	88.00	
Boucherit et al. (2022)	89.88	
FV-EffResNet	98.40	
Training 5 Testing 1	Gumusbas et al. (2019)	100.0	
Boucherit et al. (2022)	99.48	
Huang & Guo (2020)	99.53	
Liu et al. (2022)	98.11	
Ma, Wang & Hu (2023)	99.82	
FV-EffResNet	98.45	

Table 9 Accuracy comparison of proposed FV-EffResNet method with other methods using NUPT-FV.

Train/Test split	Method	Accuracy (%)	
Training 6, Testing 4	Guo et al. (2022) *	95.41	
	FV-EffResNet	98.48	
Notes.

* Results presented based on multi- modal fusion of finger print and finger vein.

Figure 5 CMC curves of the proposed FV-EffResNet using different train-test data split (A) FV_USM (B) MMCBNU_600 (C) SDUMLA (D) NUPT-FV.

As the MMCBNU_600 and SDUMLA datasets contain images obtained in a single session, we present, in both cases, comparisons based on the various train-test split ratios employed in the existing literature. As shown in Table 7, three comparisons are made when considering the MMCBNU_600, In a scenario where for each subject, training with five and testing with five, our method exhibits better performance than Gumusbas et al. (2019), showing a 2% improvement. When the training set is increased to six and testing with four, our method could not outperform (Liu et al., 2022) but still achieves superior performance compared to Gumusbas et al. (2019). Lastly, employing eight for training and two for testing, our method could not match the 100% reported by Gumusbas et al. (2019). Similarly, for SDUMLA, three split ratios are compared, as presented in Table 8, in the scenario where four images are utilized for training and two for testing, our method outperforms the compared literature. However, in the other two settings, it was marginally surpassed by the compared methods. In the case of NUPT-FV, we applied a similar approach to that of Guo et al. (2022) utilizing only the data from the first session. The dataset was divided into six training and four testing images for each subject. Our approach achieved higher accuracy despite solely utilizing the finger vein modality, in contrast to Guo et al. (2022), which incorporated both finger vein and fingerprint modalities, as presented in Table 9.

Finger vein verification

Finger vein verification involves a procedure of evaluating whether two compared images are from the same subject. If the two images are evaluated to belong to the same identity, the match is categorized as genuine; otherwise, it is considered an imposter match. In this context, the equal error rate (EER) and the receiver operating characteristic (ROC) curve are employed as evaluation metrics. ROC curves depict the performance of binary classification, illustrating the relationship between the false acceptance rate (FAR) and the genuine acceptance rate (GAR). EER represents the FAR value at the point on the ROC curve where both FAR and GAR are equal. FAR and GAR can be expressed as follows:

(2) FAR=Number of misclassified negative samplesNumber of negative samples

(3) GAR=Number of correctly classified positive samplesNumber of positive samples

As noted earlier, this study specifically concentrates on close-set evaluation, where the class labels of finger vein images in any database are known. In the context of test image I, images from different classes are designated as ‘negative samples,’ while images from the same class are categorized as ‘positive samples.’ If the predicted label of a negative sample aligns with the label of image I, it is considered a misclassified negative sample. Conversely, if the predicted label of a positive sample matches the label of image I, it is considered a correctly classified positive sample. The comparison results presented are based on the six-four train test split ratio for MMCBNU_600, and first session NUPT_FV dataset, four-two split ratio for the SDUMLA dataset, and eight-four for FV_USM two sessions combined. EER results of our method and other methods are compared in Table 10. Moreover, in Fig. 6, the ROC curves depict the performance of our network across each of the utilized databases. As presented in, Table 10, on the MMCBNU_600 dataset, although our method is outperformed by Tang et al. (2019), Huang et al. (2021) and Huang et al. (2023), it still achieved a lower EER compared to Hao, Fang & Yang (2020) and Yang et al. (2020). On the FV_USM dataset, our method achieved the third-lowest EER, despite being outperformed by Huang et al. (2021), Huang et al. (2023). Notably, our method attained the lowest EER on the NUPT_FV dataset. It is worth noting that although some of the compared methods may have outperformed our proposed method, ours still maintains superiority in terms of parameter count, being the lowest among them.

Table 10 EER comparison of proposed FV-EffResNet with other considered methods.

		Databases	
Methods	FV_USM	MMCBNU_600	SDUMLA	NUPT-FV	
Tang et al. (2019)	0.50	0.17	0.87	–	
Hao, Fang & Yang (2020)	0.74	0.29	1.17	–	
Yang et al. (2020)	0.95	1.11	1.71	–	
Boucherit et al. (2022)	1.37	–	0.99	–	
Huang et al. (2021)	0.34	0.08	0.35	–	
Huang et al. (2023)	0.20	0.18	1.10	–	
FV-EffResNet	0.39	0.21	0.43	0.18	

Figure 6 ROC curves of the proposed FV-EffResNet (A) FV_USM (B) MMCBNU_600 (C) SDUMLA (D) NUPT-FV.

Network execution time and parameter count comparison

Tables 11 and 12 present a comparison of parameter counts and processing times between the proposed network and other deep learning models respectively. The mean execution time represents the average processing time required to evaluate a single image input from the benchmark databases under consideration. It is important to highlight that our comparison of network parameter counts includes literature explicitly stating this information with the exception of Tamang & Kim (2022) and Boucherit et al. (2022) where we implemented their algorithms based on the specifications provided. As indicated in Table 11, the proposed network boasts the lowest parameter count compared to other methods, implying a more compact and resource-efficient model. Also as shown in Table 12, although the network suggested by (Ma, Wang & Hu, 2023) is marginally faster than ours, our proposed network outperforms the one suggested by Chai et al. (2022) by a significant margin in terms of processing speed.

Discussion

This study aims to provide improvements in the design and performance of deep networks for finger vein recognition. To this end, we developed an efficient CNN network named FV-EffResNet by exploring three main network design parameters. First, our network employed a domain-specific CNN block EffRes, which has significantly fewer parameters compared to other methods in the literature, rectangular kernels are employed in the proposed method, implemented through two consecutive layers using n × 1 and 1 × m convolutions, as opposed to the conventional n × m convolution. Importantly, experiments have shown that this choice of rectangular kernels is found to be more suitable for handling finger vein data. Through the incorporation of the proposed EffRes block, we have achieved a lightweight architecture with significantly fewer parameters in comparison to those put forth in the existing literature.

Table 11 Parameter count comparison of the proposed method and other method comparison.

Method	No of parameters	
Boucherit et al. (2022)	1.85 M	
Chai et al. (2022)	3.35 M	
Tamang & Kim (2022)	26.72 M	
Huang et al. (2023)	1.40 M	
Liu et al. (2022)	3.51 M	
Ma, Wang & Hu (2023)	8.10 M	
FV-EffResNet	1.32 M	

Table 12 Algorithm execution time comparison of the proposed FV-EffResNet with other method.

Methods	Inference execution time	
Chai et al. (2022)	10.44 ms	
Ma, Wang & Hu (2023)	4.00 ms	
FV-EffResNet	5.60 ms	

Secondly, the integration of the Swish activation function specifically in the hidden layers of networks dedicated to finger vein recognition has been shown here to offer unique benefits. This is because it allows for a small number of negative weights to be propagated through, which may encourage better information flow in challenging datasets like finger vein images, making it highly suitable for processing such data. Lastly, we accomplished faster convergence by implementing the concept of cyclical learning, which not only facilitated estimating the most efficient learning rate for training the architecture but also improved the efficiency and effectiveness of the network in handling finger vein image data. These results highlight the effectiveness of our method in achieving state-of-the-art performance while minimizing computational requirements, thereby contributing to the practical implementation of finger vein recognition systems.

Conclusions

In this study, we have introduced an efficient lightweight CNN based on the proposed EffRes block. The design of the EffRes block was meticulously crafted to boost the efficiency of feature extraction of finger vein features while concurrently minimizing the parameter count. This reduction in parameters significantly alleviates the overall computational burden of the system, resulting in a lightweight model. Additionally, to improve the network’s learning capability, we adopted the Swish function as activation in the hidden layers of the proposed network which we hypothesized and verified to be more suited in networks for finger vein images. Furthermore, the proposed model’s training incorporates an efficient and effective learning strategy called the cyclical learning method to enhance the network’s convergence rate and properties, resulting in improved generalization performance.

To assess the performance of our proposed network, extensive experiments were conducted on four publicly available finger vein datasets. The obtained results were then compared to the state-of-the-art approaches, which confirmed that the proposed system is highly efficient and lightweight. Consequently, the findings indicate that the system is more practical and resource-efficient, making it suitable for real-time applications. While the proposed FV-EffResNet demonstrates commendable performance in both finger vein identification and verification, there are still certain issues that warrant further investigation. Firstly, the current performance of the proposed method is based on a closed-set setting, future work could explore open-set evaluations to enhance the model’s generalization capabilities. Additionally, extending the current network to incorporate multiple tasks such as spoof detection represents a crucial avenue for future research.

Supplemental Information

Supplemental Information 1 Python codes for the proposed FV-EffResNet

Supplemental Information 2 Python codes to execute the experiments

Additional Information and Declarations

Competing Interests

Author Contributions

Data Availability

The authors declare there are no competing interests.

Yusuf Suleiman Tahir performed the experiments, analyzed the data, performed the computation work, prepared figures and/or tables, and approved the final draft.

Bakhtiar Affendi Rosdi conceived and designed the experiments, analyzed the data, authored or reviewed drafts of the article, and approved the final draft.

The following information was supplied regarding data availability:

The Python codes for our newly developed CNN-based deep learning architecture are available in the Supplemental File.

The Finger vein images databases are available at:

- FV-USM: http://drfendi.com/fv_usm_database/

- SDUMLA-HMT: https://time.sdu.edu.cn/kycg/gksjk.htm

- NUPT-FV: https://github.com/REN382333467/NUPT-FPV

- MMCBNU_6000: https://wavelab.at/sources/Drozdowski20a/.

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
