# Peer review of "FV-EffResNet: an efficient lightweight convolutional neural network for finger vein recognition"

_PeerJ Computer Science, doi:10.7717/peerj-cs.1837_

## Round 0.1 · original submission · Major Revisions

· Academic Editor

Major Revisions

The contribution of the paper needs to be clarified. Authors need to thoroughly review and address the comments made by reviewers. You need to check, refer, and compare with more recent papers. It is not a must to use the links or references mentioned by the reviewers unless they are suitable for your paper, since these should be just examples of the references you need to enrich the work with them.

·

Basic reporting

Basic reporting looks appropriate.

Experimental design

No Comments

Validity of the findings

No comments

Additional comments

It would be interesting to see how image augmentation could help in inducing environmental factors in the images. ex. rise in blood viscosity due to low temperature

Reviewer 2 ·

Basic reporting

1. General Remarks:

The article focuses on CNN-based network to accurately identify people based on finger vein pattern.
Overall, the article has a clear structure with comprehensive experimental details and comparisons with various existing methods. However, the article lacks depth in certain key areas.

2. Major Recommendations:

2.1 The article mentions the use of a square convolution kernels. Still, there isn't sufficient detail on how these work. The authors need to elucidate the specific workings of these methods and how they differ from existing ones. For instance, why was this particular mechanism chosen? How does it significantly benefit biometric recognition?

2.2 The paper code used seems limited. To enhance the reliability and generalization capability of the model, the authors might add train and test lines.

2.3 The tables showcase commendable results, but deeper analysis on key results is necessary.

2.4 The conclusion is quite concise. It might be beneficial for the authors to delve deeper into future research directions. Given the current limitations of the algorithm, what potential improvements can be explored?

3. Minor Modifications and Suggestions:


3.1 Technical finger vein recognition is used in some paragraphs which might need further elaboration to ensure complete understanding for the readers.

3.2 Labels and titles for figures and tables need to be clearer, and ensure proper citation within the text.

4. Summary:

In summary, the paper brings forth a promising method in finger vein recognition. However, it needs strengthening in depth, literature citations, and certain technical specifics. Looking forward to the authors addressing these suggestions and further refining the paper.

Experimental design

Authors proposed an approach of finger vein identification based on CNN. Therefore, they explain the validity of this system by conducting some experiments.
In my opinion, In describing the experimental setup, more details might be needed, such as specific parameters for model training and testing (execution time), etc., to aid other researchers in replicating the experiments.

Validity of the findings

Seems okay

Reviewer 3 ·

Basic reporting

The writing in the manuscript needs improvement as there are numerous mistakes that diminish its readability. The authors are advised to carefully review the manuscript and rectify all errors.
- In some sections, the authors used "finger vein," while in other parts, they used "finger-vein." Consistency in terminology is essential, and the authors should use the same format throughout the manuscript.
- In line 41, why does the word "near" start with a capital letter in "Near-infrared imaging techniques"? A similar inconsistency is observed with "finger-vein" in line 47. Additionally, in lines 97 and 99, there is inconsistency in capitalization with "Capsule networks" and "Capsule Networks."
- In line 111, there is an error in the citation. The authors wrote "Tingting et al. in their work (Chai et al., 2022)." However, they mentioned Tingting et al., but the citation is for Chai et al.
- In formal writing, such as in papers, it is common to use words instead of numerals for expressing numbers. Therefore, in lines 113 to 116, it is preferable for the authors to write: "To validate their hypothesis, a shallow fully convolutional network was designed, comprising five convolutional layers, four max-pooling layers, one per-feature LRN layer, one batch normalization layer, one LReLU layer, one fully connected layer, and a SoftMax layer responsible for generating classification probabilities."
- In line 127, it is customary to first introduce the full form of "DL" along with its abbreviation before using the abbreviation alone.

Authors should use papers published in 2023; most of the references cited are outdated. Dr. Wenxiong Kang has numerous works published in top journals, particularly in the field of finger vein research, and his database is freely available. The authors should actively engage with and compare their method to his contributions. I strongly recommend incorporating his work into your paper and reporting experiments using his database.
https://www.scholat.com/auwxkang.en
https://scholar.google.com/citations?user=meU7EOAAAAAJ&hl=en

The authors should consider utilizing additional databases and conducting comparisons with works published in 2022 and 2023.
- They should create plots for identification rate vs. rank and accuracy/loss vs. the number of epochs.
- The authors asserted that their method is a finger-vein identification approach that achieves a balance between network size, speed, and accuracy. To support this claim, the authors should incorporate a table detailing the time and memory usage for each step of their method. It is crucial to provide information on the duration of network training and the subsequent processing and verification/identification times.

Experimental design

no comment

Validity of the findings

What distinguishes the authors' work from others? The EffNet blocks, Swish activation, and cyclic learning rate technique were proposed by other researchers; the authors, however, have incorporated and adapted these methods into their own work.

Additional comments

In lines 35 and 36, the authors assert, "On the other hand, intrinsic modalities are hidden beneath the skin. As such, they are more difficult to fake." The claim regarding increased difficulty needs substantiation and should be supported with a reference. It is crucial to avoid making absolute statements about the superiority of a biometric trait like finger vein over others without context. The effectiveness of a biometric trait depends on its specific use case; for example, in forensics, fingerprint, palmprint, and DNA are commonly utilized, while for unlocking a cell phone, face and fingerprint recognition are typically employed.

---

## Round 0.2 · accepted · Accept

· Academic Editor

Accept

Thanks to the authors for having the reviewers' comments carefully considered.
Congratulation!

Reviewer 2 ·

Basic reporting

The author has address most of my earlier comments. The approach can be accepted for publication

Experimental design

Sufficient

Validity of the findings

Can be further augmented by comparing with state of the art methods

Reviewer 3 ·

Basic reporting

no comment

Experimental design

no comment

Validity of the findings

no comment

Additional comments

no comment